# THE TILTED VARIATIONAL AUTOENCODER: IMPROVING OUT-OF-DISTRIBUTION DETECTION

**Griffin Floto**
University of Toronto
`griffin.floto@mail.utoronto.ca`

**Stefan Kremer, Mihai Nica**
University of Guelph
`{skremer,nicam}@uoguelph.ca`

## ABSTRACT

A problem with using the Gaussian distribution as a prior for a variational autoencoder (VAE) is that the set on which Gaussians have high probability density is small as the latent dimension increases. This is an issue because VAEs aim to achieve both a high likelihood with respect to a prior distribution and at the same time, separation between points for better reconstruction. Therefore, a small volume in the high-density region of the prior is problematic because it restricts the separation of latent points. To address this, we propose a simple generalization of the Gaussian distribution, the tilted Gaussian, whose maximum probability density occurs on a sphere instead of a single point. The tilted Gaussian has exponentially more volume in high-density regions than the standard Gaussian as a function of the distribution dimension. We empirically demonstrate that this simple change in the prior distribution improves VAE performance on the task of detecting unsupervised out-of-distribution (OOD) samples. We also introduce a new OOD testing procedure, called the Will-It-Move test, where the tilted Gaussian achieves remarkable OOD performance.

## 1 INTRODUCTION

Due to its simplicity, the Gaussian distribution is a common prior for the variational autoencoder (VAE) (Kingma & Welling, 2014; Rezende et al., 2014). One drawback it has is that the region of high probability density becomes relatively smaller as the latent dimension increases. To see why this is an issue, consider the objective of the VAE. It tries to encode points such that they are close to the prior and can reconstruct points into their original form. Given a limited capacity of an encoder/decoder model, points in the latent space must be separated to have significant differences in their reconstructed points. With a sufficiently complex data set, it would be required to have a large volume in the high density region of the prior distribution to accommodate each of the latent points, while allowing for sufficient separation.

We argue that the Gaussian distribution's volume under regions of high probability density is not large enough to accommodate real data sets. To this end, we show that many of the points encoded by Gaussian prior VAEs exist in low-density regions, and that the high-density region remains relatively empty. In support, Nalisnick et al. (2019a) report that the latent point at the highest density of a Gaussian VAE trained on MINST was an all-black image. To deal with these issues, we propose a simple generalization of the Gaussian distribution called the tilted Gaussian distribution. We create this distribution by "exponentially tilting" the ordinary multivariate Gaussian distribution by its norm. The operation of exponential tilting is a common procedure in such diverse fields as statistical mechanics, large deviations or importance sampling, but we believe using it for VAEs as we do here is a novel contribution. The tilted Gaussian has a maximum probability density lying on the surface of a sphere rather than at a single point. A single parameter corresponds to the sphere's radius, allowing for control of the volume under the high-density region of the distribution. We show that the tilted Gaussian has exponentially more volume that the standard Gaussian as as function of the latent dimension, allowing for a far greater proportion of points from a dataset to exist in regions of

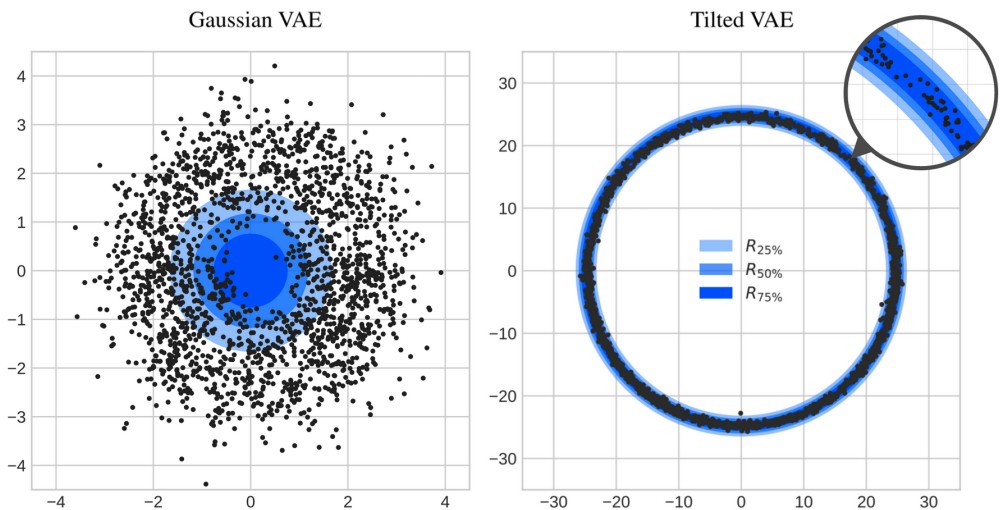

Figure 1: A 2D representation of the 10D latent space of the Gaussian VAE vs the tilted VAE with $\tau = 25$ trained on the Fashion-MNIST dataset, plotted with an *isoradial* projection that preserves the radius (i.e. $r_{2D} = \|z_{10D}\|$; see Appendix F.1 for details on this isoradial projection). Shaded regions indicate where the latent probability density is at least 25%, 50% or 75% respectively of its maximum, $R_c := \{z \mid \rho(z) \geq c \max_w \rho(w)\}$. The encoded points of the tilted VAE lie almost entirely in the region of high probability density, while the ordinary Gaussian VAE places points outside these regions. See also Section 3.5 for more comparisons.

high probability density. We investigate a simpler method of increasing the prior volume by using a Gaussian with large variance, however the effect on performance was minimal D.2.

To demonstrate the benefits of the tilted Gaussian as a prior for the VAE, we focus on the task of OOD detection. It has been noted that somewhat surprisingly, VAEs assign high likelihood to OOD points, despite being optimized on a lower bound of the log-likelihood (Nalisnick et al., 2019a; Choi et al., 2019). A possible contributing factor to this poor performance is that the high-density region of the latent space is not densely populated by in-distribution (ID) points due to the volume considerations previously detailed. We show that VAEs using the tilted Gaussian as a prior (which we call the tilted VAE), have a far greater percentage of points in high density regions (see Figure 1 for an illustration and Section 3.5 for detailed numbers) and perform significantly better on the OOD task (See Table 2). While the improvement is a step towards robust OOD detection with VAEs, we show that a prior distribution alone cannot achieve the desired level of performance on this task. Thus, we propose a new test, called the Will-It-Move test, for the OOD problem. Combined with the tilted Gaussian as a prior, it consistently improves the performance of current methods to perform OOD detection with VAEs. (See Section 4.2 for a description and Table 2 for results)

## 2 RELATED WORK

### 2.1 EXTENSIONS OF VARIATIONAL AUTOENCODERS

In this section, we give a non-exhaustive list of extensions proposed for VAEs. The majority of approaches aim to increase the flexibility of the prior. For example, a mixture of Gaussians can be used as an alternative (Dilokthanakul et al., 2016) to the standard Gaussian prior. The VampPrior attempts to improve upon the mixture of Gaussians by using a mixture of variational posteriors (Tomczak & Welling, 2018). Another proposal for the prior distribution is the hyperspherical VAE (Davidson et al., 2018), which uses a von-Mises-Fisher (VMF) distribution. In contrast, our tilted prior is concentrated around the hypersphere as a "soft" constraint, which allows the ordinary normal distribution instead of the VMF to be used. Other proposed methods use the Dirichlet process (Nalisnick & Smyth, 2017), the Chinese restaurant process (Goyal et al., 2017), and the Gaussian

process (Casale et al., 2018). Hierarchical constructions of priors can be used to construct prior distributions as in Sønderby et al. (2016), Maaløe et al. (2016), and Maaløe et al. (2019), using various methods to construct the latent variable hierarchy. A different approach is to enforce a lower bound on the Kullback–Leibler divergence (KLD) term of the VAE, taken by the delta-VAE (Razavi et al., 2019). A powerful way to achieve greater flexibility is to use normalizing flows, a class of invertible transformations that can be used to construct more complex posteriors.

## 2.2 OUT-OF-DISTRIBUTION DETECTION METHODS FOR VARIATIONAL AUTOENCODERS

Arguably the simplest approach to perform OOD detection with VAEs is by using a one-sided test based on the likelihood that VAEs assign to data points (Bishop, 1994). Given that this method performs surprisingly poorly on the OOD task Nalisnick et al. (2019a), a number of alternative scores have been provided. Likelihood Ratios (Ren et al., 2019) proposes to use the ratio between two different types of models, one capturing the semantic content of data, the other capturing background information. Likelihood Regret (Xiao et al., 2020) uses a similar principle, but uses the ratio between an model that is optimized for the training dataset and another optimized for an individual sample. ROSE (Choi et al., 2021) uses a method to compute how much a sample would update a model's parameters. Input complexity (Serrà et al., 2020) use an estimate of the Kolmogorov complexity as well as the log-likelihood estimate assigned by the VAE. Density of States Estimation (Morningstar et al., 2021) uses an approach based on statistical physics and directly measures the typicality of different model statistics to classify OOD samples. Nalisnick et al. (2019b) uses a typicality test and Song et al. (2019) uses batch normalization statistics. Ran et al. (2021) takes a different approach by adding Gaussian noise to images and employs a noise contrastive prior to the VAE architecture.

# 3 THE TILTED GAUSSIAN DISTRIBUTION AND THE TILTED VAE

## 3.1 REVIEW OF THE VARIATIONAL BOUND USED FOR TRAINING VAEs

Consider a dataset $X = \{x^i\}_{i=1}^N$ consisting of $N$ i.i.d. samples of some distribution in $\mathbb{R}^{d_x}$. We model the data by a two-step process: first, an unobserved latent variable $z \in \mathbb{R}^{d_z}$ is sampled according to a prior distribution $p_Z(z)$; second, a value $x^i$ is produced conditionally on the latent $z$ by a parameterized generator ("decoder") $p_\theta(x|z)$ according to some unknown parameter $\theta = \theta^*$. To attempt to recover $\theta^*$, we maximize the marginal log-likelihood of the data $\sum_{i=1}^N \log p_\theta(x^i) = \sum_{i=1}^N \int_{\mathbb{R}^{d_z}} \log p_\theta(x^i|z) p_Z(z) dz$. As the integral over the latent variables is often intractable, the variational lower bound is used to introduce a parameterized inference model ("encoder") $q_\phi(z|x)$ to approximate the true posterior $p_\theta(z|x)$. Given an encoder model $q_\phi$, the *variational lower bound* of the log-likelihood is

$$\sum_{i=1}^N \log p_\theta\left(x^i\right) \geq \sum_{i=1}^N \mathbb{E}_{q_\phi(z|x^i)}\left[\log p_\theta\left(x^i|z\right)\right] - \sum_{i=1}^N D_{\mathrm{KL}}(q_\phi(z|x^i) \| p_Z(z)), \qquad (1)$$

where $D_{\mathrm{KL}}(\cdot)$ is the Kullback-Leibler divergence (KLD). The KLD term in the equation above can be interpreted as fitting the aggregated posterior $q_\phi(z|x^i)$ to the pre-determined prior $p_Z(z)$. Typically, the standard Gaussian distribution $p_Z(z) \sim \mathcal{N}(0, I)$ is used as the prior and the encoder distribution is $q_\phi(z|x) \sim \mathcal{N}\left(\mu(x), \mathrm{diag}(\sigma^2(x))\right)$ with parameterized functions $\mu(x), \sigma^2(x) \in \mathbb{R}^{d_z}$ for the encoder mean and variance. In this case, the KLD term appearing in (1) can be evaluated explicitly as

$$D_{\mathrm{KL}}(\mathcal{N}\left(\mu, \mathrm{diag}(\sigma^2)\right) \| \mathcal{N}(0, I)) = \frac{1}{2}\|\mu\|^2 + \frac{1}{2}\sum_{j=1}^{d_z}\left(\sigma_j^2 - 1 - \log \sigma_j^2\right). \qquad (2)$$

this formula makes the lower bound of (1) explicit, and can be optimized using stochastic gradient descent. Gaussian noise is added to each point during training so that the latent point $z^i$ assigned to the data point $x^i$ is $z^i = \mu(x^i) + \sigma(x^i)\mathcal{N}(0, I)$, see e.g. (Kingma & Welling, 2014).

## 3.2 IMPROVING THE VAE

The primary technical contribution of this paper is the proposal of an alternative prior distribution $p_Z(z)$, the exponentially tilted Gaussian prior. As the standard Gaussian prior has maximum density

at the origin, the KLD term in the log-likelihood bound (2) forces all latent points into the same location, $\mu_i = 0$ and $\sigma_i = 1$. This leads to "crowding" around the origin, making it difficult to differentiate between encoded data samples (Hoffman & Johnson, 2016; Alemi et al., 2018).

Unlike the standard Gaussian, the tilted prior does not have maximum density at a single point. Instead, the maximum density occurs at all the points on the hyper-sphere of radius $\tau$ as illustrated in Figure 2. This allows the model to spread out latent points while still optimizing the marginal log-likelihood bound (1).

Additionally, the radii $\|z\|$ of points drawn from this distribution are near $\tau$ with high probability.[1] Therefore, the norm of an encoded point $\|z\|$ drawn from $z \sim q_\phi(z|x)$ can form a simple statistic which is an effective test for OOD points (see (7)). We also obtain a new variational bound, the analogue of equation 2, for the KLD of the tilted Gaussian distribution. This allows training the tilted VAE by a simple modification to the standard Gaussian. This new prior can substitute for the standard Gaussian with minimal changes to existing code. In practice, the only change is replacing the term $\frac{1}{2}\|\mu\|^2$ in the KLD from equation (2) to $\frac{1}{2}\left(\|\mu\| - \|\mu^*\|\right)^2$ where $\|\mu^*\|$ is a fixed constant depending on the tilting parameter $\tau$. This corresponds to the KLD of the new tilted model as in equation (6).

### 3.3 EXPONENTIALLY TILTED GAUSSIAN DISTRIBUTION AND THE TILTED VAE

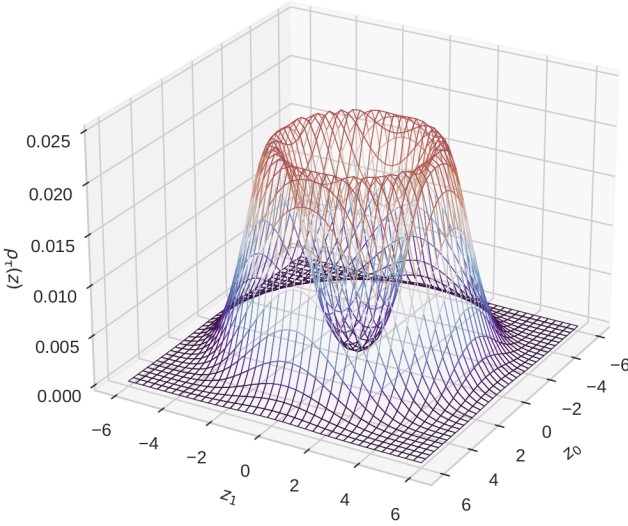

Figure 2: Example of $\rho_\tau(z)$ when $d_z = 2$, $\tau = 3$

**Definition 3.1.** For a tilting parameter $\tau \geq 0$, the *exponentially tilted Gaussian distribution*, denoted $\mathcal{N}_\tau(0, I)$, is the random variable on $\mathbb{R}^{d_z}$ with probability density $\rho_\tau(z)$ defined by:

$$\rho_\tau(z) := \frac{e^{\tau\|z\|}}{Z_\tau} \frac{e^{-\frac{1}{2}\|z\|^2}}{\sqrt{2\pi}^{d_z}} = \frac{e^{\tau\|z\|}}{Z_\tau} \rho_0(z), \quad Z_\tau := \mathbb{E}_{z \sim \mathcal{N}(0,I)}[e^{\tau\|z\|}] \tag{3}$$

Compared to the standard Gaussian, which corresponds to $\tau = 0$, the tilting term $e^{\tau\|z\|}$ pushes the distribution towards values greater than $\|z\|$. By completing the square, one sees the density is proportional to $e^{-\frac{1}{2}(\|z\|-\tau)^2}$, meaning that the density is radially symmetric and has a maximum value at $\|z\| = \tau$. An illustration is provided in Figure 2.

The tilted VAE uses the tilted Gaussian as the latent prior $p_Z(z)$. An important feature of the tilted VAE is that the encoder distribution (i.e. the distribution assigned in latent space to a single input $x_i$)

---

[1]For a standard Gaussian, $\|z\|$ is concentrated around $\sqrt{d}$ as $d \to \infty$, so the effect of tilting pushing $\|z\|$ toward $\tau$ will only have an appreciable effect when $\tau > O(\sqrt{d})$; see Figure 5 (Right) for an illustration.

is *not* exponentially tilted. We take this distribution to be a simple Gaussian of the form $z \sim \mathcal{N}(\mu, I)$ where $\mu = \mu(x)$ is determined by the encoder parameters. For the sake of simplicity and to reduce computation we have fixed the covariance matrix $\Sigma = I$, whereas for the Gaussian VAE the variance $\Sigma = \mathrm{diag}(\sigma)$ is allowed to be chosen by the encoder. We believe that it would be possible to also allow this extra flexibility for the tilted VAE but we do not do so here.[2]

Intuitively, the tilted prior allows the encoder distributions to choose $\mu$ anywhere on the surface of a hyper-sphere with a minimum KLD, rather than located at the single point $\mu = 0$, as would be the case when the prior is an standard Gaussian. Note also that since a Gaussian cannot be perfectly fit to the distribution $\mathcal{N}_\tau(0, I)$, there will always be a non-zero minimum KLD between the encoder and prior distributions, i.e. $\delta(\tau) := \inf_{\mu \in \mathbb{R}^{d_z}} D_{\mathrm{KL}}(\mathcal{N}(\mu, I) \parallel \mathcal{N}_\tau(0, I)) > 0$ when $\tau > 0$. This can be interpreted as the minimum average amount of information in nats that each sample contains after being encoded. The decoder is then able to use this information to differentiate distributions in the latent space. This minimum value $\delta(\tau)$ is referred to as the "committed rate" in Razavi et al. (2019).

### 3.4    Normalization Constant and the Tilted KLD Bound

The calculation of the distribution's normalization constant $Z_\tau$ and $D_{\mathrm{KL}}(\mathcal{N}(\mu, I) \parallel \mathcal{N}_\tau(0, I))$ is summarized below with proofs deferred to Appendix A. The normalizing constant $Z_\tau$ satisfies

$$Z_\tau = M\left(\frac{d_z}{2}, \frac{1}{2}, \frac{1}{2}\tau^2\right) + \tau\sqrt{2}\frac{\Gamma(\frac{d_z+1}{2})}{\Gamma(\frac{d_z}{2})}M\left(\frac{d_z+1}{2}, \frac{3}{2}, \frac{1}{2}\tau^2\right), \tag{4}$$

where $M$ is the Kummer confluent hypergeometric function $M(a, b, z) = \sum_{n=0}^{\infty} \frac{a^{(n)}z^n}{b^{(n)}n!}$ and $a^{(n)}$ is the rising factorial $a^{(n)} = a(a+1)\ldots(a+n-1)$. The KLD can then be written as

$$D_{\mathrm{KL}}(\mathcal{N}(\mu, I) \parallel \mathcal{N}_\tau(0, I)). = \log Z_\tau - \tau\sqrt{\frac{\pi}{2}}L_{\frac{1}{2}}^{\frac{d_z}{2}-1}\left(-\frac{\|\mu\|^2}{2}\right) + \frac{\|\mu\|^2}{2}, \tag{5}$$

where $L$ is the generalized Laguerre polynomial, $L_n^{(\alpha)}(x) = \binom{n+\alpha}{n}M(-n, \alpha+1, x)$. During training, instead of computing $D_{\mathrm{KL}}(\mathcal{N}(\mu, I) \parallel \mathcal{N}_\tau(0, I))$ exactly (involving the difficult-to-compute Laguerre polynomial in (5)), we use the following simple quadratic approximation of the KLD, vastly simplifying the computation while preserving the bound (1):

$$D_{\mathrm{KL}}(\mathcal{N}(\mu, I) \parallel \mathcal{N}_\tau(0, I)) \leq \frac{1}{2}\left(\|\mu\| - \|\mu^\star(\tau)\|\right)^2 + C^*(\tau) \tag{6}$$

where $\mu^\star(\tau) = \mathrm{argmin}_\mu D_{\mathrm{KL}}(\mathcal{N}(\mu, I) \parallel \mathcal{N}_\tau(0, I))$ and $C^*(\tau) = \min_\mu D_{\mathrm{KL}}(\mathcal{N}(\mu, I) \parallel \mathcal{N}_\tau(0, I))$ is the minimum such value[3]. This bound allows easy training of the tilted VAE just as (2) allowed training of the standard Gaussian VAE. Note that the term $C^*(\tau)$ is constant and can therefore be omitted during training. The proof of (6) is deferred to Appendix A.3. Figure 5 illustrates the difference between the exact vs approximate KLD in (6) and shows how $\|\mu^\star(\tau)\|$ depends on $d_z$.

### 3.5    Comparison Between Tilted Gaussian and Regular Gaussian

We now highlight some important differences between the tilted and regular Gaussian distribution as the dimension of the space grows. First, the volume of the space under regions of high probability density is much larger for the tilted Gaussian. Second, regions of high probability density have a much larger contribution to the total probability for the tilted Gaussian over the regular Gaussian.

To demonstrate these facts, consider a given density function $\rho(z)$ and $c \in [0, 1]$. We are interested in the set of points that have a density at least $c$ times the maximum possible density. This set is denoted $R_c := \{z \mid \rho(z) \geq c \max_{w \in \mathbb{R}^{d_z}} \rho(w)\}$. In Appendix B, we show that the volume of $R_c$ for the tilted Gaussian grows exponentially faster than the regular Gaussian with respect to the dimension of the space. The contribution to the total probability by the set $R_c$ is given by $\mathbb{P}_\rho[R_c] := \int_{z \in R_c} \rho(z)\mathrm{d}z$.

---

[2]Note that whenever we use the standard Gaussian VAE in this paper (e.g. to compare to the tilted VAE), we *do* allow the variance $\sigma$ to be chosen as a parameter

[3]Note that $\mu^\star(\tau)$ is not unique and all vectors lying on the hyper-sphere with radius $\|\mu^\star\|$ are valid minima.

For both the tilted and regular Gaussian distribution, the integral cannot be solved in closed-form. Appendix B shows a comparison between the distributions when $c = 0.5$.

Empirically, we observe that when using the tilted VAE, a far greater proportion of latent points are embedded in regions of high probability density. Figure 1 shows an example of the differences in latent spaces when using the two different priors.

| REGION $R_c := \{z : \rho(z) > c \max_w \rho(w)\}$ | STANDARD GAUSSIAN | | | TILTED GAUSSIAN ($\tau = 25$) | | |
|---|---|---|---|---|---|---|
| | % POINTS | PROB. | VOL. | % POINTS | PROB. | VOL. |
| $R_{25\%}$ | 21.3 % | 1.4% | 417 | 99.9% | 88.5% | $3.4 \times 10^{14}$ |
| $R_{50\%}$ | 6.8% | 0.8% | 13.1 | 99.5% | 73.5% | $2.3 \times 10^{14}$ |
| $R_{75\%}$ | 1.2% | 0.01% | 0.2 | 95.2% | 52.7% | $1.5 \times 10^{14}$ |

Table 1: Numerical comparison of the "size" of high probability regions for the standard Gaussian distribution and tilted Gaussian distribution in 10D. The rows correspond to the regions $R_{25\%}, R_{50\%}, R_{75\%}$ where the probability density is at least 25%, 50% or 75% of the maximum probability density, $R_c = \{z : \rho(z) > c \max_w \rho(w)\}$ as illustrated in Figure 1. Three separate notions of "size" are compared as follows. "% POINTS": The fraction of the data set the VAE in Figure 1 has in the region. "PROB." : The probability $\mathbb{P}[z \in R_c]$ when $z$ is chosen according to the latent distribution, " VOL.": The volume of the region.

## 4 OUT-OF-DISTRIBUTION DETECTION

### 4.1 USING THE LIKELIHOOD AS AN OOD SCORE

We now focus on a particular application of VAEs to demonstrate the advantages of using the tilted Gaussian as a prior. The task of OOD detection can be described by considering a training distribution $P$. Samples that have a high probability density under the training distribution are considered to be ID, whereas samples that have a low probability density under the training distribution are considered to be OOD. In practice it is common to use data from a different distribution $Q$ as examples of OOD data. As we only have access to in-distribution training data, the OOD task we described is a case of positive unlabelled (PU) learning Jaskie & Spanias (2022) Given that the VAE is optimized to get a lower bound on the marginal log-likelihood, (1) is a natural statistic to determine if a sample is OOD. Samples with a higher marginal log-likelihood would be considered to be more likely to be ID. In this work we assume that the reconstruction or likelihood term of (1) takes the form of a Gaussian likelihood and we approximate the expectation over a single sample $q_\phi(z|x^i)$. This allows the marginal log-likelihood bound to be used as an "OOD score" namely:

$$\mathcal{S}_{\text{OOD}}(x) := -\|x - \hat{x}\|^2 - \frac{1}{2} \left(\|z\| - \|\mu^\star(\tau)\|\right)^2. \tag{7}$$

where $z \sim q_\phi(z|x)$ is a sample of the latent space image of the point $x$, and $\hat{x}$ is the reconstructed data point from the image $z$, i.e. $\hat{x} \sim p_\theta(x|z)$. We note that value of $C^\star(\tau)$ in the KLD approximation (6) is not required as it is constant for all inputs. In practice, some value of $\mathcal{S}_{\text{OOD}}(x)$ must be used as a threshold to determine if a point is OOD or not. By sweeping through all possible threshold values we obtain the full ROC curve.

We conduct an experiment comparing the tilted Gaussian vs the standard Gaussian as a prior for the VAE on two OOD detection tasks. To give a quantitative measure of OOD classification, we examine the Receiver Operating Characteristics (ROC) curve looking at the relationship between the false positive and true positive rates as the threshold value for (7) changes. The Area Under the Curve-Receiver Operating Characteristics (AUCROC) (Fawcett, 2006) is used to give a single number measuring OOD detection performance. Details regarding the full experimental settings can be found in Appendix C and the results are shown in Figure 3[4].

---

[4]Code is available at `https://github.com/anonconfsubaccount/tilted_prior`

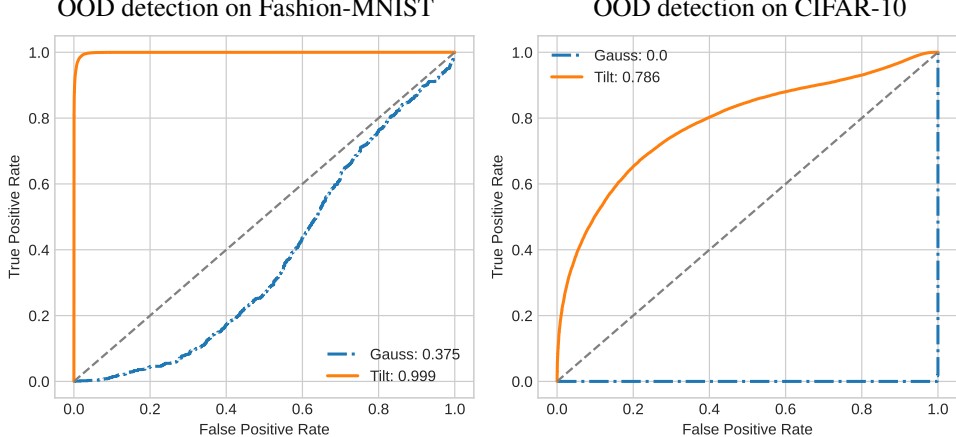

Figure 3: A comparison of the ROC of VAEs trained with a standard Gaussian prior and an tilted prior. The left plot corresponds to models trained on the Fashion-MNIST and CIFAR-10 data sets, respectively. In both cases, the MNIST data set is used as OOD data. The AUCROC metrics are shown in the plot legends. In both cases the tilted Gaussian prior leads to a significant increase in performance of the OOD detection task.

## 4.2 THE WILL-IT-MOVE TASK FOR OUT-OF-DISTRIBUTION DETECTION

### 4.2.1 MOTIVATION: ENTROPIC ISSUES WITH VARIATIONAL AUTOENCODER

As detailed in Caterini & Loaiza-Ganem (2022), using the marginal likelihood from the VAE as an OOD score has fundamental issues for low-entropy data distributions. To describe this issue theoretically, imagine we are given samples from a "true" distribution $P$ to distinguish from an "outside" distribution $Q$. We train a parameterized model $p_\theta$ to maximize the log-likelihood of points from $P$, i.e. we maximize $\mathcal{L}_P(\theta) = \mathbb{E}_{X\sim P}[\log p_\theta(X)]$ [5]. Once $p_\theta$ is trained, the lower bound of $\log p_\theta(x)$ is used as an OOD score; points with low log-likelihood are to be labelled from $Q$ rather than $P$ because one would hope that $\mathcal{L}_P(\theta) > \mathcal{L}_Q(\theta)$ for our choice of parameters.

However, this simple log-likelihood comparison can fail when the entropy of the "outside" distribution $Q$ is low compared to that of $P$. One can decompose the log-likelihood into two terms:

$$\mathcal{L}_P(\theta) = \mathbb{E}_{X\sim P}[\log p_\theta(X)] = -D_{\mathrm{KL}}(P \parallel p_\theta) - \mathbb{H}[P],$$

where $\mathbb{H}[P]$ is the entropy of the distribution $P$. This decomposition shows that comparing the expected log-likelihood of $P$ to that of $Q$ gives

$$\mathcal{L}_P(\theta) < \mathcal{L}_Q(\theta) \iff \mathbb{H}[P] - \mathbb{H}[Q] > D_{\mathrm{KL}}(Q \parallel p_\theta) - D_{\mathrm{KL}}(P \parallel p_\theta).$$

Even if our model $\theta$ is perfectly chosen so that $D_{\mathrm{KL}}(P \parallel P_\theta)) = 0$, it is possible to observe a higher average log-likelihood for points from the "outsider" distribution $Q$ when $\mathbb{H}[P] > \mathbb{H}[Q] + D_{\mathrm{KL}}(Q \parallel P_\theta)$. This phenomenon means that distributions $Q$ with low entropy, (i.e. $\mathbb{H}[Q]$ small) can sometimes be hard to detect using a VAE model. Indeed, experiments show that detecting a constant image as out of distribution can counter-intuitively be quite difficult (see "Constant" row in Table 2). To deal with this problem, we propose the Will-It-Move testing procedure.

### 4.2.2 DESCRIPTION OF THE WILL-IT-MOVE TEST

The Will-It-Move (WIM) test works by fine-tuning the parameters of the VAE to try to "push" OOD latent points away from the model prior, while keeping known ID points close to their original prior. Points that are truly OOD should be easily moved, while points from the original training

---

[5]Note that in practice, one maximizes the empirical average over a given sample, but here we will focus on the theoretical average that one gets in the large data limit: $\lim_{n\to\infty} \frac{1}{n}\sum_{i=0}^{n} \log p_\theta(x_i) = \mathbb{E}_{X\sim P}[\log p_\theta(X)]$

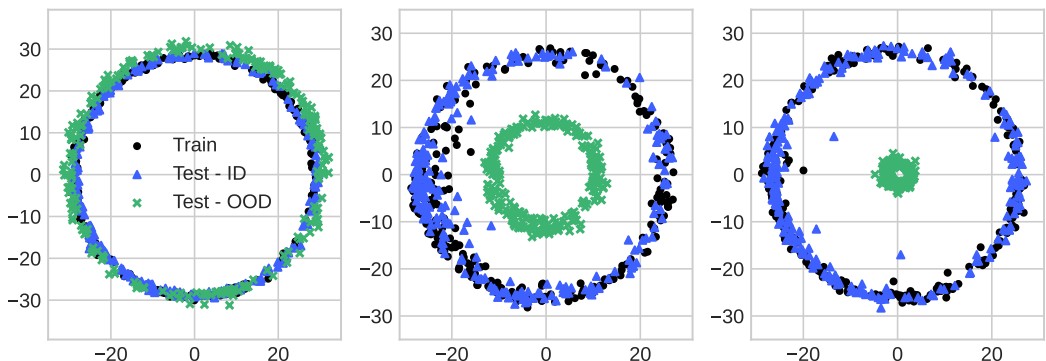

Figure 4: A 2D representation of the 10D latent space of the Tilted Gaussian VAE during the WIM test with CIFAR-10 as the ID and MNIST as the OOD. *Isoradial* projection is used (i.e. $r_{2D} = \|z_{10D}\|$ see Appendix F.1 for details). Black circles are from the training set, while green x and purple triangles are unknown points to be classified from either OOD distribution or ID respectively. (Note that the true identity of the test points is *unknown* to the algorithm.) Originally, the VAE trained only on the training point maps all points to the latent distribution well. During parameter fine-tuning in the WIM test, only the truly OOD points are easily moved away to the alternative latent distribution (in this case, a standard Gaussian) and can therefore be easily identified.

distribution should be hard to move away from the original prior. In other words, the answer to "Will-It-Move?" is "Yes" for OOD points and "No" for ID points during the test. After the WIM-fine-tuning is completed, the same score (7) is then used to label unknown points as either OOD or ID. By moving points away form the model prior, this allows the KLD term in the likelihood bound to become very large and overcome the entropy problem previously discussed.

To precisely describe the WIM task, consider a training data set $X$ drawn iid from a distribution $P$. We suppose that we have already used $X$ to train a VAE by maximizing the log-likelihood w.r.t. some given latent distribution $\mathcal{Z}_X$. (e.g. using the tilted Gaussian $\mathcal{Z}_X = \mathcal{N}_\tau(0, I)$). The WIM test fine-tunes the parameters of the VAE to label the identity of points from an "unknown" data set $U$ that has samples from both the training and the OOD distribution i.e. $U = \{x^i\}_{i=1}^n \cup \{\tilde{x}^i\}_{j=1}^m$ where $x^i \sim P$ and $\tilde{x}^j \sim Q$. Note that the identity of the points in $U$ is *not* known apriori.

To do this, we choose *a different* latent space distribution for the data $U$, denoted by $\mathcal{Z}_U$. In the WIM test, the model fine-tunes the parameters so that points from $X$ stay close to $\mathcal{Z}_X$ and points from $U$ move to $\mathcal{Z}_U$ in latent space. Specifically, we maximize by gradient descent the objective function that is the weighted sum of their log-likelihoods for some $\alpha \in \mathbb{R}$:

$$\mathcal{L}(\mathcal{Z}_X, X) + \alpha \mathcal{L}(\mathcal{Z}_U, U) \tag{8}$$

where $\mathcal{L}(\mathcal{Z}_X, X) := \sum_{x^i \in X} \mathbb{E}_{q_\phi(z|x^i)} \left[ \log p_\theta \left( x^i | z \right) \right] - D_{\mathrm{KL}}(q_\phi(z|x^i) \parallel \mathcal{Z}_X))$ and analogously $\mathcal{L}(\mathcal{Z}_U, U)$ is the log-likelihood for the data set of unknown points $U$ w.r.t. the alternative latent distribution $\mathcal{Z}_U$.

The first term of (8) is the original training objective of the VAE, and is intended to keep the data set $X$ in it's original configuration. The effect of the second term is to push the points of $U$ toward $\mathcal{Z}_U$ in latent space. The hope is that OOD points from $Q$ can easily be moved to $\mathcal{Z}_U$, whereas points from $P$ are similar enough to the original data set so that the first term in (8) keeps these points around $\mathcal{Z}_X$ despite the pull from the second term. The points that we observe moving during this fine-tuning are therefore identified as the OOD points (hence the name "Will-It-Move?").

In practice, we use $\mathcal{Z}_X$ as a tilted Gaussian and $\mathcal{Z}_U$ as a standard Gaussian. Implementation details of the WIM test are discussed in D.1. As an ablation test, we perform the WIM test with both $\mathcal{Z}_X$ and $\mathcal{Z}_U$ as Gaussian distributions with different location parameters and observed significantly better performance with the tilted Gaussian. Results from this experiment can be found in D.2

## 5 RESULTS AND COMPARISON TO OTHER OOD METHODS

Table 2: AUROC comparison between OOD detection methods with Fashion-MNIST and CIFAR-10 as training distributions for various OOD sets. A larger number is better. Note that when AUROC < 0.5, (indicated by ∗), then "flipping labels" would improve the classifier; see F.2 for discussion.

| DATASET | GAUSS | IC (PNG) | IC (JPEG2000) | RATIO | REGRET | TILT | WIM |
|---|---|---|---|---|---|---|---|
| FASHION-MNIST | | | | | | | |
| MNIST | 0.375* | 0.993 | 0.351* | 0.965 | 0.999 | 0.999 | **1.0** |
| CIFAR-10 | **1.0** | 0.970 | **1.0** | 0.914 | 0.986 | 0.997 | **1.0** |
| SVHN | **1.0** | 0.999 | **1.0** | 0.761 | 0.989 | 0.98 | **1.0** |
| KMNIST | 0.765 | 0.863 | 0.769 | 0.960 | 0.998 | 0.999 | **1.0** |
| NOISE | **1.0** | 0.324* | **1.0** | **1.0** | 0.998 | **1.0** | **1.0** |
| CONSTANT | 0.975 | **1.0** | 0.984 | 0.980 | 0.999 | 0.798 | **1.0** |
| | | | | | | | |
| CIFAR-10 | | | | | | | |
| MNIST | 0.0* | 0.976 | 0.0* | 0.032* | 0.986 | 0.797 | **1.0** |
| FASHION-MNIST | 0.032* | 0.987 | 0.035* | 0.335* | 0.976 | 0.688 | **1.0** |
| SVHN | 0.209* | 0.938 | 0.215* | 0.732 | 0.912 | 0.143* | **0.991** |
| LSUN | 0.833 | 0.348* | 0.833 | 0.508 | 0.606 | 0.933 | **0.941** |
| CELEBA | 0.676 | 0.310* | 0.679 | 0.404* | 0.738 | 0.877 | **0.997** |
| NOISE | **1.0** | 0.042* | **1.0** | 0.851 | 0.994 | **1.0** | **1.0** |
| CONSTANT | 0.015* | **1.0** | 0.269* | 0.902 | 0.974 | 0.0* | **1.0** |

To further validate the performance of our proposed prior, we compare the tilted prior to a variety of methods that achieve top performance in OOD detection with VAEs. The methods we consider are Input complexity (**IC (png)**), (**IC (JPEG2000)**) (Serrà et al., 2020), likelihood ratios (**Ratio**) (Ren et al., 2019), and likelihood regret (**Regret**) (Xiao et al., 2020). A VAE with a standard Gaussian prior and log-likelihood OOD score (**Gauss**) is also used as a benchmark. The tilted VAE and WIM test are denoted **Tilt** and **WIM** respectively. Table 2 is a summary of the results from this analysis.

From this experiment, we observe that the tilted VAE significantly outperforms the Gaussian VAE across a variety of datasets. Furthermore, the tilted VAE alone has competitive performance with the OOD based scores we compare against. When the WIM test is used with the tilted prior we observe that this method matches or improves upon the compared methods in all tests considered. For difficult cases such as when CIFAR-10 is the training distribution and the OOD datasets are SVHN, LSUN and CELEBA the WIM test achieves significant performance increases over all compared methods. When looking at cases where the OOD dataset is much simpler that the training distribution, for example the constant dataset, we observe methods that use the log-likelihood estimate as an OOD score (**Gauss** and **Tilt**) perform poorly.

## 6 CONCLUSION

We propose a generalization of the Gaussian distribution called the tilted Gaussian, and show that it can be implemented in the same way as the standard Gaussian VAE with a simple change to the KLD term. It's use as a prior distribution for the VAE was motivated by showing that when the standard Guassian is used as a prior, a large percentage of data points are encoded into regions of low probability density. We then prove that the tilted Gaussian has exponentially more volume in high probability density regions that the standard Gaussian as a function of the distribution dimension. Empirically, we show that the tilted VAE encodes a far greater percentage of points on regions of high-probability density. We introduce a new OOD score for the VAE and empirically demonstrate that it is a consistent improvement over the performance of recent OOD scores for the VAE. Finally, we perform an ablation study and show that the WIM test performs better when using the tilted Gaussian prior over the standard Gaussian prior.

While this work investigated the tilted Gaussian distribution in the context of OOD detection with VAEs, we believe that there are many applications where the tilted Gaussian is a viable drop-in replacement for the standard Gaussian. We encourage researchers to experiment with this distribution in other situations.

## ACKNOWLEDGEMENTS

We thank 3 anonymous reviewers for their thorough reading of the paper, and for the many suggestions which improved the paper.

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

## A  FACTS ABOUT THE TILTED GAUSSIAN

### A.1  DERIVATION OF THE NORMALIZATION CONSTANT $Z_\tau$

$$\rho_\tau(z) = \frac{e^{\tau\|z\|}}{Z_\tau} \frac{e^{-\frac{1}{2}\|z\|^2}}{\sqrt{2\pi}^d} dz$$

$$Z_\tau = \mathbb{E}_{z\sim\mathcal{N}(0,I)}\left[e^{\tau\|z\|}\right] = \int_{\mathbb{R}^d} e^{\tau\|z\|} \frac{e^{-\frac{1}{2}\|z\|^2}}{\sqrt{2\pi}^d} \mathrm{d}z$$

$$= \mathbb{E}_{x\sim\chi(d)}\left[e^{\tau x}\right] = \int_0^\infty e^{\tau x} \frac{x^{d-1}e^{-\frac{1}{2}x^2}}{2^{\frac{1}{2}d-1}\Gamma\left(\frac{d}{2}\right)} \mathrm{d}x \text{ where } \chi(d) \overset{d}{=} \|z\|$$

$$= \sum_{n=0}^\infty \frac{\tau^n \mathbb{E}\left[\chi^n\right]}{n!}$$

$$= \sum_{n \text{ even}}^\infty \frac{\tau^n d(d+2)\dots(d+n-2)}{n!} + \sum_{n \text{ odd}}^\infty \frac{\tau^n \mu_1 (d+1)(d+3)\dots(d+n-2)}{n!}$$

$$\text{where } \mu_1 = \mathbb{E}\left[\chi\right] = \sqrt{2}\frac{\Gamma\left(\frac{d+1}{2}\right)}{\Gamma\left(\frac{d}{2}\right)}$$

$$= M\left(\frac{d}{2}, \frac{1}{2}, \frac{1}{2}\tau^2\right) + \tau\sqrt{2}\frac{\Gamma\left(\frac{d+1}{2}\right)}{\Gamma\left(\frac{d}{2}\right)} M\left(\frac{d+1}{2}, \frac{3}{2}, \frac{1}{2}\tau^2\right)$$

where we have used the moments of the $\chi(d)$ distribution and we let $M\left(a, b, z\right) = \sum_{n=0}^\infty \frac{a^{(n)}}{b^{(n)}} \frac{z^n}{n!}$ is the Kummer confluent hypergeometric function and $a^{(n)} = a(a+1)\dots(a+n-1)$ is the rising factorial.

### A.2  EXACT FORMULA FOR THE KLD USED IN THE VAE

Define the shorthand $f_\tau(\mu)$ to be the KLD defined below:

$$f_\tau(\mu) := D_{\text{KL}}(\mathcal{N}(\mu, I) \| \mathcal{N}_\tau(0, I)) \tag{9}$$

$$= \mathbb{E}_{z\sim\mathcal{N}(\mu,I)}\left[\ln\left(\frac{\rho_{\mathcal{N}(\mu,I)}(z)}{\rho_{\mathcal{N}_\tau(0,I)}(z)}\right)\right]$$

$$= \mathbb{E}_{z\sim\mathcal{N}(\mu,I)}\left[\ln\left(\frac{e^{-\frac{1}{2}\|z-\mu\|^2}}{\frac{1}{Z_\tau}e^{\tau\|z\|}e^{-\frac{1}{2}\|z\|^2}}\right)\right]$$

$$= \mathbb{E}_{z\sim\mathcal{N}(\mu,I)}\left[\ln\left(Z_\tau e^{-\tau\|z\|}e^{-\frac{1}{2}\|z\|^2+\langle z,\mu\rangle-\frac{1}{2}\|\mu\|^2+\frac{1}{2}\|z\|^2}\right)\right]$$

$$= \ln(Z_\tau) - \tau\mathbb{E}_{z\sim\mathcal{N}(\mu,I)}\left[\|z\|\right] + \mathbb{E}_{z\sim\mathcal{N}(\mu,I)}\left[\langle z,\mu\rangle\right] - \frac{1}{2}\|\mu\|^2$$

$$= \ln(Z_\tau) - \tau\mathbb{E}_{z\sim\mathcal{N}(\mu,I)}\left[\|z\|\right] + \frac{1}{2}\|\mu\|^2$$

$$= \ln(Z_\tau) - \tau L_{\frac{1}{2}}^{(d/2-1)}\left(-\frac{\|\mu\|^2}{2}\right) + \frac{1}{2}\|\mu\|^2,$$

where we have used $\mathbb{E}_{z\sim\mathcal{N}(\mu,I)}\left[\|z\|\right] = \mathbb{E}[\chi_\mu] = L_{\frac{1}{2}}^{(d/2-1)}\left(-\frac{\|\mu\|^2}{2}\right)$ and $L_n^{(\alpha)}$ is the generalized Laguerre polynomial.

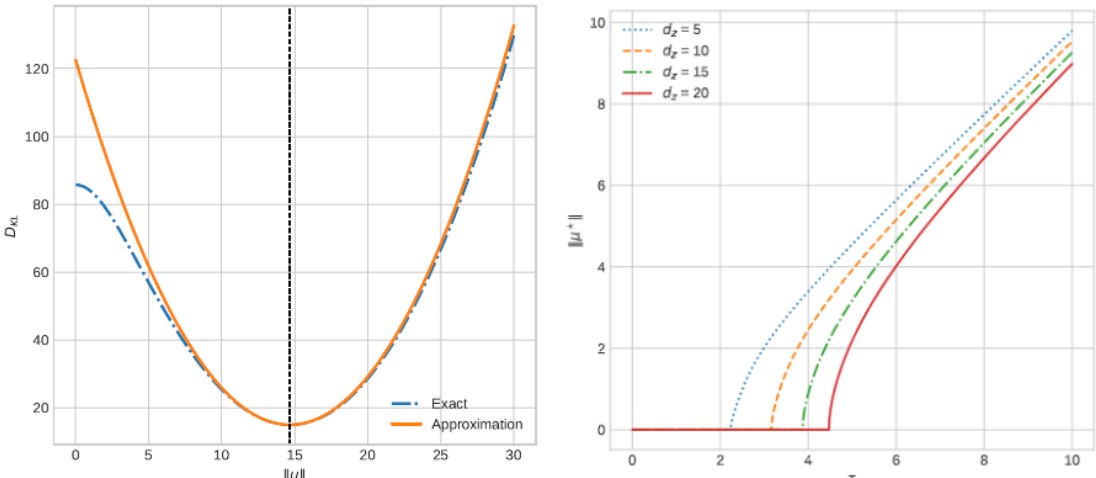

Figure 5: Left: Comparison between the exact and quadratic approximation of the KLD $D_{\mathrm{KL}}(\mathcal{N}(\mu, I) \| \mathcal{N}_\tau(0, I))$ as in (6) for $d_z = 10, \tau = 15$ Right: $\|\mu^\star(\tau)\|$ plotted for a variety of different latent dimensions $d_z$. The value of $\|\mu^\star(\tau)\|$ was computed numerically using gradient descent from the explicit formula in (5). Note that there is a critical $\tau$ value for which $\mu^*(\tau)$ becomes nonzero. We conjecture that the critical $\tau \approx \sqrt{d}$ as $d \to \infty$.

### A.3 Approximation of the KLD

Let $f_\tau(\mu) \in \mathbb{R}$ be the KLD as in (9). Note that $f_\tau$ depends only on the magnitude of $\mu$, not the direction. With $x = \|\mu\|$, we can therefore define $g_\tau : \mathbb{R} \to \mathbb{R}$ by

$$g_\tau(x) := f_\tau(x\vec{e}_1) = \ln(Z_\tau) - \tau \mathbb{E}_{z \sim \mathcal{N}(0,I)}\left[\|z + x\vec{e}_1\|\right] + \frac{1}{2}x^2 \tag{10}$$

where $\vec{e}_1$ is the unit vector $\vec{e}_1 = (1, 0, 0, \ldots, 0)^T$.

**Proposition A.1.** *The function $g$ satisfies*

$$g_\tau''(x) = 1 - \tau \mathbb{E}_{z \sim \mathcal{N}(0,I)}\left[\frac{z_2^2 + \ldots + z_d^2}{\|z + x\vec{e}_1\|^3}\right]$$

*and in particular for $\tau > 0$,*

$$g_\tau''(x) \leq 1$$

*Proof.* The formula for $g''$ goes by taking the derivative of the definition $g_\tau(x)$ from (10) using $\frac{\mathrm{d}^2}{\mathrm{d}x^2}\frac{1}{2}x^2 = 1$ and the elementary fact

$$\frac{\mathrm{d}^2}{\mathrm{d}x^2}\|z + x\vec{e}_1\| = \frac{z_2^2 + \ldots + z_d^2}{\|z + x\vec{e}_1\|^3} \geq 0,$$

Since this is non-negative, it is immediate that $g_\tau''(x) \leq 1$ as claimed.

$\square$

*Remark A.2.* Note that the random variable appearing in the formula for $g_\tau$ is related to the non-central Beta distribution

$$X\left(\frac{m}{2}, \frac{n}{2}, \lambda\right) \stackrel{d}{=} \frac{\sum_{i=n+1}^{n+m} z_i^2}{(z_1 + \lambda)^2 + \sum_{i=2}^{m} z_i^2 + \sum_{j=n+1}^{n+m} z_j^2} \in (0, 1)$$

**Corollary A.3.** *Let $\mu^*(\tau)$ be the minimum of $f_\tau$ so that $f_\tau(\mu^*(\tau)) \leq f_\tau(\mu)$ for all $\mu \in \mathbb{R}^d$ and $g_\tau(\|\mu^*(\tau)\|) \leq g_\tau(x)$ for all $x \in \mathbb{R}$. Then:*

$$g_\tau(x) \leq \frac{1}{2}\left(x - \|\mu^*(\tau)\|\right)^2 + g_\tau\left(\|\mu^*(\tau)\|\right)$$

*or equivalently:*

$$f_\tau(\mu) \leq \frac{1}{2}\left(\|\mu\| - \|\mu^*(\tau)\|\right)^2 + g_\tau\left(\|\mu^*(\tau)\|\right)$$

*Proof.* Since $g_\tau''(x) \leq 1$ and $g$ is differentiable, by Fermat's theorem, $g_\tau'(\|\mu^*(\tau)\|) = 0$. We can now integrate to find that for $x > \|\mu^*(\tau)\|$

$$g_\tau'(x) = g_\tau'(\|\mu^*(\tau)\|) + \int_{\|\mu^*(\tau)\|}^{x} g_\tau''(w)\,\mathrm{d}w$$

$$\leq 0 + \int_{\|\mu^*(\tau)\|}^{x} 1\,\mathrm{d}w$$

$$= x - \|\mu^*(\tau)\|$$

and finally integrating again gives the desired inequality for $x > \|\mu^*(\tau)\|$:

$$g_\tau(x) = g_\tau\left(\|\mu^*(\tau)\|\right) + \int_{\|\mu^*(\tau)\|}^{x} g_\tau'(w)\,\mathrm{d}w$$

$$\leq g_\tau\left(\|\mu^*(\tau)\|\right) + \int_{\|\mu^*(\tau)\|}^{x} \left(w - \|\mu^*(\tau)\|\right)\,\mathrm{d}w$$

$$= g_\tau\left(\|\mu^*(\tau)\|\right) + \frac{1}{2}(x - \|\mu^*(\tau)\|)^2$$

The same inequality holds for $x < \|\mu^*(\tau)\|$ by integrating from $\int_x^{\|\mu^*(\tau)\|}$. The final inequality for $f$ follows by setting $x = \|\mu\|$. $\qquad\square$

## B  REGIONS OF HIGH PROBABILITY DENSITY FOR GAUSSIAN VS TILTED

To compute the "size" of regions where the probability density function is at least $c \in (0,1)$ times its maximum, $R_c = \{\rho(z) > c \max_w \rho(w)\}$, we compute for the standard Gaussian that:

$$\rho(z) \geq c\rho(0) \iff e^{-\frac{1}{2}\|z\|^2} \geq c \iff \|z\| \leq \sqrt{2\ln\left(\frac{1}{c}\right)}.$$

This shows that this region is always a sphere for the standard Gaussian, $R_c = B_{\sqrt{2\ln(c^{-1})}}(0)$. In contrast, for the tilted Gaussian we have:

$$\rho_\tau(z) \geq c\rho_\tau(\tau) \iff e^{-\frac{1}{2}(\|z\|-\tau)^2} \geq c \iff \tau - \sqrt{2\ln\left(\frac{1}{c}\right)} \leq \|z\| \leq \tau + \sqrt{2\ln\left(\frac{1}{c}\right)}$$

This shows that the region of high probability is a "shell", the difference of two spheres $R_c = B_{\tau+\sqrt{2\ln(c^{-1})}}(0) \setminus B_{\tau-\sqrt{2\ln(c^{-1})}}(0)$ [6]. The formula for the volume of the high dimensional sphere $V_d(r) = \frac{\pi^{d/2}}{\Gamma(\frac{d}{2}+1)}r^d$ from which we can compute volumes:

$$Vol_{\text{Gaussian}}(R_c) = \frac{\pi^{d/2}}{\Gamma(d/2+1)}\left(2\ln(1/c)\right)^{d/2}$$

---

[6]Note that the inner sphere is trivial unless $\tau > \sqrt{2\ln(1/c)}$

$$Vol_{\text{Tilted}}(R_c) = \frac{\pi^{d/2}}{\Gamma(d/2+1)} \left( \left(\tau + \sqrt{2\ln(1/c)}\right)^d - \left(\tau - \sqrt{2\ln(1/c)}\right)^d \right)$$

By the "curse/blessing of dimensionality", the region of high probability density is exponentially larger for the tilted prior as a function of $d$. We compute the probability of landing in the region $R_{50\%}$ by numerically integrating the probability density on these regions below.

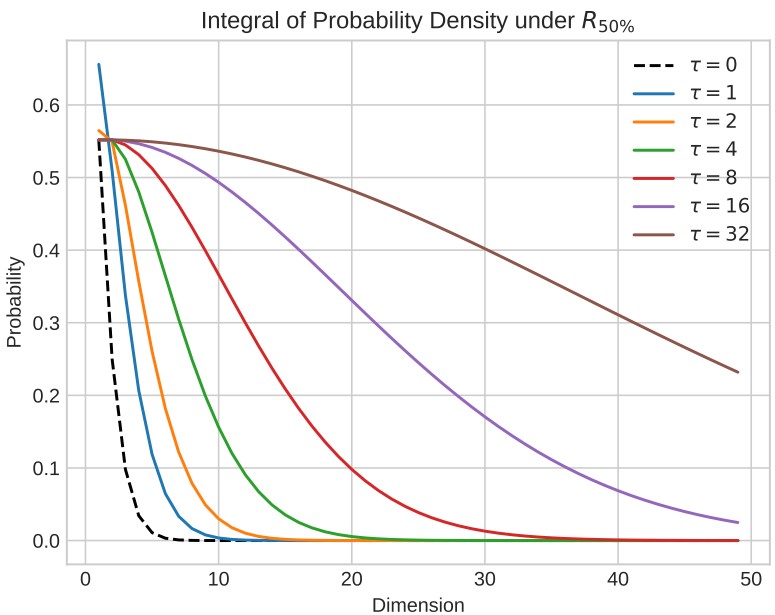

Figure 6: The probability $\mathbb{P}[\mathcal{N}_\tau(0, I) \in R_{50\%}]$ i.e. the contribution of $R_{50\%}$ to the total probability. Note that $\tau = 0$ corresponds to the standard Gaussian.

## C  EXPERIMENTAL SETTINGS

### C.1  MODEL STRUCTURE AND OPTIMIZATION PROCEDURE

The parameters used for the tilted prior are $\tau = 30$ for the Fashion-MNIST test and $\tau = 25$ for the CIFAR-10 test. The WIM test uses $\alpha = 0.1$ for all of experiments. For all tests we train the VAE for 250 epochs with a batch size of 64. We use the ADAM optimizer with a learning rate of $10^{-4}$ and clip gradients that are greater than 100. The encoder and decoder of the VAE is based on the work done by (Choi et al., 2019) and consists of 5 convolutions layers with a fully connected layer for both $\mu$ and $\sigma$. The decoder uses 6 deconvolutional layers and ends with a single convolutional layer. The use of a fully connected encoder layers for the tilted prior is essential as the KLD is a function of $\|\mu\|$ which is impractical to optimize with a fully convolutional model. We initialize all convolutional weights from the distribution $\mathcal{N}(0, 0.2)$. The model details can be viewing in C.1. On the Fashion-MNIST test we use latent dimensions $d_z = 10$ and for the CIFAR-10 test we used $d_z = 100$. Both Likelihood Regret and Likelihood Ratio are intended to be run with categorical cross entropy loss, thus we employ this as the reconstruction loss function for all comparison models.

### C.2  IMPLEMENTING METHODS

For all comparison methods we use IWAE with 200 samples to derive a lower bound on the log-likelihood. For likelihood regret, we use gradient descent on all model parameters for 100 steps in each of the tests. We use the ADAM optimizer for this process at a learning rate of 1e-4, same as the

Table 3: Details of the model architecture where $b$ is the batch size, $f$ is the number of convolutional filters, and $w$ is the width of the square image. After each layer of the encoder and decoder a leaky rectified linear unit was used. The filter parameter was set to $f = 32$ for all experiments.

| ENCODER | DECODER |
| --- | --- |
| INPUT $x$ | INPUT $z$, RESHAPE TO $b \times nz \times 1 \times 1$ |
| $5 \times 5$ CONV$_{c \times f}$, STRIDE 1 | $w/4 \times w/4$ DECONV$_{d_z \times 2f}$, STRIDE 1 |
| $5 \times 5$ CONV$_{f \times f}$, STRIDE 2 | $5 \times 5$ DECONV$_{2f \times 2f}$, STRIDE 1 |
| $5 \times 5$ CONV$_{f \times 2f}$, STRIDE 1 | $5 \times 5$ DECONV$_{2f \times f}$, STRIDE 2 |
| $5 \times 5$ CONV$_{2f \times 2f}$, STRIDE 2 | $5 \times 5$ DECONV$_{f \times f}$, STRIDE 1 |
| RESHAPE TO $b \times d'_z$ | $5 \times 5$ DECONV$_{f \times f}$, STRIDE 2 |
| LINEAR FROM $d'_z$ TO $d_z$ FOR $\mu$ AND $\sigma$ | $5 \times 5$ CONV$_{f \times c}$, STRIDE 1 |

training procedure. We train the background model in Likelihood Ratio by setting the perturbation ratio parameter $\mu$ to be 0.2 for all tests. Besides this, the background model is trained in an identical format to the regular models. For input complexity we use the OpenCV implementations of the PNG and JP2 compression algorithms.

## C.3 DATASETS

The datasets that are used in our experiments are MNIST, Fashion-MNIST, KMNIST, CIFAR-10, SVHN, CelebA and LSUN. We also use two synthetic datasets that we call Noise and Constant. Images from the Noise dataset are created by sampling from the uniform distribution in the range of $[0, 255]$ for each pixel in an image, where the constant dataset is created by sampling from the uniform distribution in the range of $[0, 255]$ where all pixels in the image have the same value. All images are resized to shape 32 x 32. Color images are converted to gray-scale for the Fashion-MNIST test by taking the first channel and discarding the other two. Grayscale images are converted to color images by taking 3 copies of the first channel. When testing with the CelebA and LSUN datasets, we use 50000 random samples from each due to the large dataset sizes.

# D WILL-IT-MOVE TEST DETAILS

## D.1 IMPLEMENTATION DETAILS OF THE WIM TEST

The WIM test was implemented by using batches that had equal parts training data, in-distribution test data, and OOD data. (We used 256 images of each type.) Our experiments used the full in-distribution test-set, which was equivalent to approximately $25\%$ of the OOD dataset. We trained 5 epochs on each batch, then tested both datasets for OOD points. This means that the model enjoys better performance on OOD points that it hasn't be tuned on, and shows effective generalization. The gradient of the training data was weighted more strongly by choosing $\alpha = 0.1$.

In time-sensitive applications, it could be infeasible to run the 5 epoch WIM updates on every data point given that the backward pass through the model is far slower than a forward pass. Instead, one could fine tune once only once on some data, and then run the fine tuned model for OOD detection. In table 4 we compare the relative number of images per second of the methods used in this paper. For the WIM test we include the time for the model to do 5 epochs of fine tuning with a final forward pass as well as the time for a single forward pass, which could be performed after the model has already been fine tuned.

## D.2 ABLATION STUDY

We perform an ablation of the WIM test, by replacing the tilted Gaussian prior with a standard Gaussian prior, meaning that both $\mathcal{Z}_X$ and $\mathcal{Z}_U$ are Gaussian distributions. To keep the distributions separate in the latent space, we set each element of the location parameter for $\mathcal{Z}_U$ to be 3. We observe that the tilted prior consistently improves the performance of the WIM test over the standard Gaussian prior.

Table 4: Relative runtime comparison between methods on the Fashion-MNIST dataset. A larger number is better. Experiments were run on an NVIDIA 3090 GPU with a Ryzen 3800X CPU

| METHOD | IMAGES PER SECOND |
|---|---|
| WIM - INFERENCE WITH PRE-FINE TUNED MODEL | 97.3 |
| WIM - RUNNING 5 EPOCHS OF FINE TUNING PLUS INFERENCE | 0.498 |
| IC (PNG) | 95.2 |
| IC (JP2) | 92.6 |
| RATIO | 49.0 |
| REGRET | 2.20 |

Table 5: An ablation study comparing the performance of the WIM test when a VAE is trained with a Gaussian prior VAE rather than a tilted VAE. The task is OOD detection with Fashion-MNIST and CIFAR-10 as training datasets. The metric reported is the AUCROC, where larger number is better.

| DATASET | GAUSSIAN | TILTED GAUSSIAN |
|---|---|---|
| FASHION-MNIST | | |
| MNIST | **1.0** | **1.0** |
| CIFAR-10 | 0.996 | **1.0** |
| SVHN | 0.997 | **1.0** |
| KMNIST | 0.999 | **1.0** |
| NOISE | **1.0** | **1.0** |
| CONSTANT | 0.998 | **1.0** |
| | | |
| CIFAR-10 | | |
| MNIST | 1.0 | 1.0 |
| FASHION-MNIST | 1.0 | 1.0 |
| SVHN | 0.940 | **0.991** |
| LSUN | 0.826 | **0.941** |
| CELEBA | 0.958 | **0.997** |
| NOISE | 1.0 | 1.0 |
| CONSTANT | 1.0 | 1.0 |

# E   COMPARISON TO LARGE VARIANCE GAUSSIAN

We investigate the result of using a standard VAE with a large variance Gaussian prior. The form of the prior is $\mathcal{N}(0, aI)$ where $a \in \mathbb{R}^+$, and we vary $a$ to understand the effect of on OOD detection.

This larger variance also increases the region of high density, but effectively increases the "scale" of the latent space. This means separating points by adding $\mathcal{N}(0, Id)$ as part of the encoder becomes less effective (i.e. Increasing the variance of the posterior $a$ is equivalent to decreasing the variance added to each point during encoding.) It is not surprising therefore that changing the variance does not improve the ordinary VAE. Results can be found in Table 6.

# F   OTHER DETAILS

## F.1   ISORADIAL PROJECTION

The isoradial projection used in Figure 1 and Figure 4 map 10-dimensional space to 2-dimensional space, $f_{\text{isoradial}} : \mathbb{R}^{10} \to \mathbb{R}^2$ so that the norms of the vectors in the domain $\mathbb{R}^{10}$ are preserved as the radius in the range $\mathbb{R}^2$, i.e. $R = ||f_{\text{isoradial}}(z)||_{2D} = ||z||_{10D}$ for all $z \in \mathbb{R}^{10}$. This requirement determines the radius for every point $z$, so the only thing left to determine is the *angle* $\theta$ in 2D polar coordinates. In Figure 1 and Figure 4 we chose the angle according to the *first two principle components* (PCA) of the given data set. That is we chose $\theta$ to be the angle exactly that of the two dimensional

Table 6: Results comparing a standard VAE with a large variance Gaussian prior. The task is OOD detection with Fashion-MNIST and CIFAR-10 as training datasets. The metric reported is the AU-CROC, where larger number is better. Note that when AUROC $< 0.5$, (indicated by $*$), then "flipping labels" would improve the classifier; see F.2 for discussion.

| DATASET | $a = 1$ | $a = 10$ | $a = 100$ | $a = 1000$ | TILTED GAUSSIAN |
|---|---|---|---|---|---|
| FASHION-MNIST | | | | | |
| MNIST | 0.0375* | 0.992 | 0.989 | 0.986 | **0.999** |
| CIFAR-10 | **1.0** | 0.985 | 0.973 | 0.977 | 0.997 |
| SVHN | **1.0** | **1.0** | **1.0** | 0.999 | 0.980 |
| KMNIST | 0.765 | 0.913 | 0.838 | 0.883 | **0.999** |
| NOISE | **1.0** | 0.379* | 0.428* | 0.488* | **1.0** |
| CONSTANT | 0.975 | **1.0** | **1.0** | **1.0** | 0.798 |
| | | | | | |
| CIFAR-10 | | | | | |
| MNIST | 0.0* | 0.005* | 0.003* | 0.004* | **0.797** |
| FASHION-MNIST | 0.032* | 0.065* | 0.056* | 0.078* | **0.688** |
| SVHN | 0.209* | **0.417*** | 0.407* | 0.355* | 0.143* |
| LSUN | 0.833 | 0.836 | 0.823 | 0.823 | **0.933** |
| CELEBA | 0.676 | 0.767 | 0.795 | 0.760 | **0.877** |
| NOISE | **1.0** | **1.0** | **1.0** | **1.0** | **1.0** |
| CONSTANT | **0.015*** | 0.006* | 0.011* | 0.007* | 0.0* |

vector $(z_{PCA1}, z_{PCA2})$ of the first two principle components of $z$, i.e. $\theta = \arctan(\frac{z_{PCA2}}{z_{PCA1}})$ where $z_{PCA1}$ is the first component of the PCA and $z_{PCA2}$ is the second component. These components are linear combinations of the original components determined from the dataset. The projection $f$ can hence be written as

$$f_{\text{isoradial}}(z) = \big(R\cos(\theta), R\sin(\theta)\big) = \left(\|z\| \frac{z_{PCA1}}{\sqrt{z_{PCA1}^2 + z_{PCA2}^2}}, \|z\| \frac{z_{PCA2}}{\sqrt{z_{PCA1}^2 + z_{PCA2}^2}}\right)$$

## F.2 DISCUSSION OF CLASSIFIERS WITH AUROC $< 0.5$

For any classifier with an AUROC $< 0.5$, a better classifier can be constructed by "flipping" the classifier (i.e. when the original classifier outputs "OOD", the new classifier will label this as in-distribution and vice versa). The AUROC of the new classifier is simply $1-$ the AUROC of the original classifier. This is a simple way to "fix" many of the poor classifiers listed in Table 2.

For clarity, we chose to only report the AUROC of the original classifier. This makes interpreting Table 2 straightforward since the exact same classifier is used in each column (as opposed to a mixture of flipped/un-flipped classifiers). Additionally, since the choice of whether or not to flip is evidently very data dependent, in practice one would not know apriori whether flipping improves the classification score (e.g. if classifying only a single data point, there would be no way to know). OOD classifiers that consistently output high AUROC scores across many different data sets (such as the WIM method) are therefore much more useful than classifiers that sometimes have to be flipped.

In the setting of OOD for VAEs, this kind of poor performance leading to AUROC $< 0.5$ has been widely reported when the OOD distribution images are much "simpler" than the in-distribution images, see e.g. Nalisnick et al. (2019a). One possible way to understand this is from an information theory point of view, which is described in Section 4.2.1.

