# OpenReview forum: "The Tilted Variational Autoencoder: Improving Out-of-Distribution Detection"
_ICLR.cc/2023/Conference — ICLR 2023 poster_

### Official Review · Reviewer_Sn18 · 2022-10-24

**Confidence:** 4
**Correctness:** 4
**Technical Novelty And Significance:** 3
**Empirical Novelty And Significance:** 4
**Recommendation:** 6

**Clarity, Quality, Novelty And Reproducibility:**

The paper is well organised and clearly written. The proposed methods for OOD detection by VAEs with tilted Gaussian priors are novel and interesting. The experiments are well designed and convincing.

**Strength And Weaknesses:**

Paper strengths:
The authors show that learning VEAs with the proposed alternative prior is feasible and provide a justified approximation for the KL-divergence between standard Gaussians and the "tilted Gaussian" prior. The OOD classification is then done by thresholding the estimated likelihood.  The authors propose a second and stronger OOD classification approach via an additional fine-tuning of an ensemble of two VAEs with different prior distributions, which is interesting (even though it introduces additional run-time cost). The experiments are convincing. They show that the first proposed method (based on estimated likelihood) is competitive and that the advanced method outperforms state of the art methods based on generative models.

Paper weaknesses:
- I find the motivation given in Section 4.2.1, which claims to explain known issues of generative models for OOD detection as a consequence of low entropy distributions of OOD data, rather not convincing. I would rather assume that the "places" to which a VAE encoder maps OOD data is not fully predictable since such data were not used in training. It might be just the "smoothness" of the learned encoder/decoder mappings, that in most cases ensures that OOD data have smaller likelihood in the resulting model.

- I would have expected an experimental comparison with known VAE models that use more general priors as e.g. VampPrior.

**Summary Of The Paper:**

The paper considers using VAEs for out of distribution (OOD) detection . The authors argue that the standard normal prior for latent variables is too concentrated to allow a proper representation of "in distribution" data and propose to use instead a distribution that is concentrated around a sphere in the latent space. The authors show in experiments that this "tilted Gaussian" prior is indeed better suited for the OOD task.  Furthermore, they  propose an even stronger approach based on fine-tuning an ensemble composed of the learned VAE and an additional VAE with a different prior on test data.

**Summary Of The Review:**

The "out of distribution" recognition task is of high practical relevance. The paper proposes interesting and novel methods for such tasks which are based on VAEs.

---

> ### Author Response · Authors · 2022-11-14
> **Response to Reviewer Sn18**
>
> We would like to thank the reviewer for their feedback. Below we detail responses to some of the specific points they brought up.
>
> **Re: Motivation in Section 4.2.1**
> We agree there are multiple ways to look at this problem and explain the poor performance of VAEs on OOD for images which are much simpler than the in-distribution sample. We do want to emphasize that Section 4.2.1 serves only to motivate why and how we originally came up with the WIM task. Our hope is that this motivation can help the reader better understand why we came up with this task.
>
> **Re: More experimental comparisons**
> Of course, more experiments would always shed more light on the strengths/weaknesses of the method. We chose in this paper to focus on the comparisons to the models presented in Table 2 instead of more complex models like VampPrior (suggested by Reviewer Sn18), and NDCC, ODIN and countless others, including other paradigms for DGMs based on GANs or normalizing flows (mentioned by Reviewer damh), because these are more complex models and beyond the scope of what we could address in the present venue. VampPrior in particular allows for some adaptable parameters in the prior to be learned from data, rather than a prior which is chosen beforehand like we are suggesting with our Tilted prior. A fair comparison perhaps would be to look at a version of VampPrior which uses tilted Gaussians in the place of ordinary Gaussians, but still otherwise uses the new innovations in VampPrior. We hope to run experiments with more complex uses of the Tilted Prior (such as comparing to things like the VampPrior) in a more comprehensive future work rather than squeezing a superficial effort into this manuscript.

---

> > ### Comment · Reviewer_Sn18 · 2022-11-18
> > **Response**
> >
> > Overall, I think that the paper presents an interesting proposal for out of distribution detection (OOD). However the method remains heuristic in parts. The response provided by the authors has not  contributed to clarifying under which assumptions it is expected to outperform other ODD methods. I therefore keep my recommendation 6 (marginally above the acceptance threshold).

---

### Official Review · Reviewer_damh · 2022-10-24

**Confidence:** 3
**Correctness:** 4
**Technical Novelty And Significance:** 4
**Empirical Novelty And Significance:** 4
**Recommendation:** 8

**Clarity, Quality, Novelty And Reproducibility:**


## Novelty

### Contribution 1: Titled normal prior

The definition of the tilted Gaussian appears new (to me)
No related work is cited, so I conclude the authors claim the definition of this distribution as a contribution.

One possibly related distribution to cite is the Skew-Normal distribution: https://en.wikipedia.org/wiki/Skew_normal_distribution

I do think that some more work could be done by the authors to situate this distribution in the broader literature of "tilted" distributions within statistics. Even defining what statisticians mean by "tilted" would be helpful.

### Contribution 2: Will it move (WIM) for OOD detection

I'm rather familiar with OOD literature, and I don't know of approaches that "fine tune" the encodings of a probabilistic model at "test time", so this seems novel enough to me. In terms of context and comparisons, I think enough related work on how to turn probabilistic models into OOD scores has been cited already.

It may be interesting to connect the WIM idea with the sub-field of positive-unlabeled learning (PU learning).


## Quality


### W1: Several AUROC scores far less than 0.5: suggest max(AUROC,1-AUROC) instead

Reporting AUROC scores below 0.5 (chance) always suggests that something odd is happening. Indeed, Fig 3 and Table 2 each show several dataset pairs where AUROC goes well below 0.5. What is probably happening is what is described in "Do Deep Generative Models know what they don't know?", where the OOD dataset images are *simpler* than the ID images (e.g. MNIST simpler than Fashion MNIST), and thus the OOD images get *higher* likelihood scores (flexible deep models can reconstruct them better), so if we threshold out low likelihood we select OOD more often than ID. This phenomena has been widely reported.

What to do about it? I suggest that what should be reported here is max(AUROC, 1.0 - AUROC). In otherwords, allow for post-hoc "flipping" of all classifier decisions if you discover it is much better for the pair of datasets under consideration. This is logically coherent (could use a small validation set if you want to be careful) and should resolve some issues with Table 2. For example, the Gauss method applied to MNIST vs CIFAR10 currently reports an AUROC of 0.0 (not bold), but in fact this means a perfect classifier can be built (and should be bolded), as long as we know to flip the prediction.

Without such changes, I worry naive readers will take the wrong messages away from some of Table 2.
It would be fine to mark such cases with an asterisk or similar, so that it's clear when flipping is applied.
It would also be fine to provide as a target dataset a mix of simpler and more complex images, to show that in "real" applications such tricks don't always work.

### W2: Scalability of Will-it-move test?

Needing to run parameter updates *at test time* for the WIT task seems expensive. Can you clarify how the runtime is for the datasets you're looking at and better discuss the pros/cons of this procedure in light of runtime concerns?

### W3: Experiments only compare to other VAE models

The current paper nicely examines results within the subfield of VAE models.
However, plenty of other work from diverse methods pursue the OOD detection problem, and the paper could perhaps have a larger impact if more effort was made to compare to other work outside of VAEs or even outside deep generative models.

For example, deep learning approaches such as:

* NDCC: Novelty Detection Consistent Classifiers (Cheng & Vasconcelos CVPR 2021)
* ODIN: (Liang, Li, and Srikant ICLR 2018)

or countless others, including other paradigms for DGMs based on GANs or normalizing flows

I don't think this is strictly necessary and I understand if it is too much effort, but I think the payoff of such experiments is that the tilted VAE's performance could be understood by a broader community. Please understand I'm less interested in asking "did you beat state of the art?" and more interested in understanding "how does this recent VAE advance fit into the big picture? how far would VAEs have to improve to compete against X?".... Without this big picture, I'm worried the impact of the paper would be to VAE-focused researchers only (though I still think that is enough to get published).



### W4: Perhaps compare to a Gaussian prior with larger variance?

To better understand the advantage of the approach, I'm interested in comparing it to a (simpler) alternative.
I'm not totally convinced this alternative would work better, but I'd appreciate the authors' thoughts.

If the claim is that the volume of "high density" needs to be large, perhaps this could be accomplished by keeping the Gaussian prior, but allowing its variance to increase (perhaps by a lot).
We fix the prior's covariance at identity for reasons of simplicity, but it could be any multiple of identity as long as the scale is fixed.

This thought is motivated by a post-hoc analysis of Fig 1.... if the "radius" of the prior were just made larger, but the encoder and decoder were held fixed, clearly more data points would lie in the high-density regions and at least some of the claimed benefits of the tilted prior could be achieved by a (simpler) Gaussian one.


## Reproducibility

### W5: Exact design choices could be more clear in paper

* What value of hyperparameter \tau is used
* How to balance minibatches of the multi-task Will-it-Move objective? esp if X is larger than U or vice versa?
* What encoder architecture is assumed?
* What decoder architecture?

Seems like some of these choices are mentioned in Supplement, but could be better foreshadowed in main paper, and supplement could offer a nicer table (not just a dense paragraph) for easy lookup.

## Clarity

Overall, I found the manuscript rather easy to follow.

One point of note is that its actually well-understood that in high dimensions, the mass (not density) of a Gaussian concentrates in a thin spherical shell at some distance from the origin, rather than near the origin.

See:
* Blog post by John Cook: https://www.johndcook.com/blog/2011/09/01/multivariate-normal-shell/
* Fig. 3 in paper by Betancourt: https://arxiv.org/pdf/1701.02434.pdf#page=8

Sec. 3.2 says that for the tilted prior, "the radii ||z|| of points drawn from this distribution are near \tau with high probability". It's useful/important to note that (surprisingly), that for some (different) radius, the same can also be said for the Gaussian. Naturally, this is just an interesting connection, not trying to say anything is at issue in the comparison of relative volumes/masses of the high-density regions in Table 1.

Minor issues:

* in Sec. 2.2, the claim that one-sided tests of the likelihood "performs surprisingly poorly" probably needs a citation
* typo at bottom of page 2: "used construct more"
* typo on page 6: "over a single of sample"



**Strength And Weaknesses:**


# Strengths

* The tilted prior is new to me and seems useful as a way to improve the volume of high-density regions
* Results in Table 2 are convincing across many dataset pairs that the Will-it-Move idea works well for OOD
* Comparisons to many other VAE-like approaches to OOD detection in Table 2 are welcome
* Volume analysis in Table 1 is compelling argument for tilted prior over standard Gaussian

# Weaknesses

There's several issues that I'd appreciate hearing about from the authors in rebuttal

* W1: Several AUROC scores far less than 0.5 seem suspicious
* W2: Runtime Scalability of Will-it-move test?
* W3: Experiments only compare to other VAE models
* W4: Perhaps compare to a Gaussian prior with larger variance?
* W5: Exact design choices could be more clear in paper

**Summary Of The Paper:**

The paper presents two related ideas on the topic of unsupervised density modeling for out-of-distribution detection. The area of specific focus within this topic is variational autoencoders (VAEs).

First, they suggest an alternative prior for the latent code z of the generative model. Instead of the standard multivariate Gaussian, they suggest that p(z) should be a *tilted* Gaussian (see Defn 3.1 and Fig 2), with one hyperparameter \tau > 0. The key idea is that in high dimensions (when d_z is large), the highest density does not occur at one point (the origin), but
instead on a d_z-1 dimensional subspace wherever the vector z has magnitude such that ||z|| = \tau.

Table 1 compares these two candidate distributions in terms of the *volume* and *probability mass* assigned to the space whose density (pointwise) is above 25/50/75% of the maximum density. The tilted Gaussian's set has much larger volume and probability mass than the usual Gaussian (e.g. at 25%, tilted mass is >88%, standard is below 2%).
An essential claim is that because the tilted prior's high-density regions are larger, there is less "conflict" (my term) in the ELBO objective between the expected likelihood and the KL between q and prior.... the KL term can afford to map example features to more diverse codes, rather than crowding the origin.

Several useful facts are presented about this distribution, which (I believe) this paper is the first to propose
* normalizing constant is given in Eq 4
* exact KL from any Gaussian to a tilted Gaussian is given in Eq 5
* upper bound on the KL from any Gaussian to a tilted Gaussian is given in Eq 6
Eq6's tractable bound makes training expedient, as the exact KL is expensive.

Second, the work proposes a new paradigm for training OOD detection called "Will it Move". The idea is to first train an unsupervised VAE (with the tilted prior) on "normal" training data. Given an unlabeled set U containing both "normal" and "OOD" samples, we can learn to classify the OOD ones by *fine-tuning* according to a weighted objective (Eq 8) that sums an ELBO using tilted prior on the normal training data with an ELBO using the standard Gaussian prior on the mixed U data. The intention is that OOD samples in U will freely move (in latent space) to be better explained by the Gaussian prior, while the ID samples in U will stay put (because they are similar enough to the normal data).

Experiments in Table 2 examine training on Fashion MNIST / CIFAR10 then distinguishing between ID samples and OOD ones from another dataset (MNIST or KMNIST ... / SVHN or LSUN or CelebA or ...). The Will-it-Move paradigm's AUROC is above 0.94 for all dataset pairs tested, while using the VAE ELBO as a OOD score in some cases gets only 0.67 for Gaussian prior and 0.88 for tilted prior.


**Summary Of The Review:**

Overall this paper has two ideas that both seem elegant and useful, especially the tilted Gaussian prior as a way to avoid KL collapse problems (all examples push to the origin in code space).  Technical novelty is high, ideas appear sound, experiments are convincing. I recommend accepting.

I do look forward to hearing the authors' thoughts during discussion period: I think I raised a few points that will improve the paper further.

---

> ### Author Response · Authors · 2022-11-14
> **Response to Reviewer damh**
>
> This is a really detailed and thorough review! Thank you so much for this really careful review; it is obvious you put a lot of time into this and we appreciate the effort. We detail responses to the individual points you brought up below.
>
> **Re: Tilted prior distribution and skew normal**
> Indeed, we believe the tilted prior we introduce here is novel. The idea of tilting (or more fully called “exponential tilting”; see https://en.wikipedia.org/wiki/Exponential_tilting) is a common idea in statistical mechanics, large deviations or importance sampling. In that language, we are tilting the multivariable Gaussian according to its norm $\Vert z\Vert$. We added a line about this contribution to our introduction to help explain the contribution and the (otherwise confusing) name “tilting”. We do not immediately see a connection to the skew normal distribution, whose PDF seems a bit more complicated, but it is possible that such a skew would have the same effect as our tilting.
>
> **Re: Discussion about the mass of Gaussian being on a thin shell of radius $\sqrt{d}$**
> This is indeed an interesting point to consider, and is related to the fact that for $\tau < \sqrt{d}$ the exponential tilting results in $\Vert\mu^\star\Vert = 0$ (see Figure 5) and essentially no appreciable effect for the tilting unless $\tau > \sqrt{d}$. Very loosely speaking, this is because the Gaussian “naturally” will have its norm $\Vert z\Vert = O(\sqrt{d})$ so pushing it via tilting to have $\Vert z\Vert \approx \tau$, will have no effect until $\tau > O(\sqrt{d})$. In this light, the advantage of tilting becomes the fact that we can control where the mass is concentrated, and by putting this at a point further than the Gaussian would naturally go, we can achieve benefits in “size” we claim for the tilted prior. We added a brief footnote about this in Section 3.2 where this concentration is discussed highlighting this interesting phenomenon.
>
>
> **Re: Scalability of WIM Test**
> A section in the appendix was included that describes the implementation details of the WIM test. This includes a discussion on how the batching was performed as well as the scalability of the test given runtime concerns. We show the runtime speed of all the methods that were compared against in the main experiment.
> We also include the following detail. For the main results, the WIM test was tuned on a subset of the OOD data and was able to generalize to the rest of the dataset. This has practical utility in terms of the runtime as the model can be fine-tuned using the WIM test, then run using just the forward pass in an efficient manner.
>
> **Re: Connection to PU Learning**
> The relation to PU learning is an interesting connection that we were unaware of! In section 4.1 where OOD detection is described we include a mention that the OOD detection task we describe is an instance of PU learning.
>
> **Re: Comparison to Large Variance Gaussian**
> Trying a larger variance Gaussian is an interesting idea! We ran a comparison of the tilted prior to a Guassian prior with very large covariance was performed and included it in the appendix. The effect of the larger covariance was minimal across all tests, even with a very large $\Sigma = 1000 I$. We are not exactly sure what the explanation here is, but we have the following possible explanation: Simply increasing the variance does not improve performance in this case because when the variance on the posterior effectively increases the scale of the problem, so the separation of points by adding a variance 1 Gaussian to each point becomes effectively weaker. In contrast, the tilted Gaussian increases the size of the region, but doesn’t change the scaling of “nearby points”.
>
> **Re: Exact Design Choices**
> A table describing the encoder and decoder architecture in more detail has been added for easy lookup. Information about the $\tau$ hyper-parameters that were used have been moved to the beginning of the experimental settings.
>
> **Re: AUROC < 0.5 issue**
> Indeed, this is an important issue that we agree should be highlighted. We marked all the scores with AUROC < 0.5 with a star as suggested, and we added a discussion about this in the appendix. For clarity, we decided to leave the reported AUROC numbers in the table as is (instead of doing 1-AUROC), so that there would be one less “moving part” that the reader needs to interpret to understand this table. We believe separating the discussion in this way makes it easy for the reader to understand and still allows them to consider this issue.

---

### Official Review · Reviewer_T18J · 2022-10-25

**Confidence:** 4
**Correctness:** 3
**Technical Novelty And Significance:** 3
**Empirical Novelty And Significance:** 3
**Recommendation:** 6

**Clarity, Quality, Novelty And Reproducibility:**

*Clarity
- How do the authors compute mu^*?
- The format of the caption for Table 1 has some problems.
- There is no description for Isoradial projection.
- In Figure 3, it is better to describe what each chart is, e.g., "the result in each step ??". Also, this figure is not described in the main text.
- typo: in the caption of Figure1, "the tiltled VAE."

*Quality
- Please see the above comments.

*Novelty
- The proposed method seems to be novel.

*Reproducibility
- Code is available on GitHub.

**Strength And Weaknesses:**

*Strength
- The model is easy to implement by changing the parameter definition in the naive VAE algorithm.
- The model is reasonable to the problem they address and simple enough as it only has a single hyperparameter.

*Weaknesses
- Although the WIM test is reasonable for their purpose, it uses samples from both the training and the OOD distribution. I wonder how we can access OOD distribution. Also, if we can have access to the OOD distribution, why don't they simply use samples from the OOD distribution itself?
- Section 2 is unclear in relation to the proposed method. Especially, Section 2.1 just lists existing methods.
- It is better to have a more detailed discussion on and comparison with the hyperspherical VAE (Davidson et al., 2018), which also uses hyperspherical.

**Summary Of The Paper:**

This paper addresses the problem that the high probability density region of the ordinary Gaussian prior becomes small as the latent dimension increases in VAE.
The authors proposed a tilted Gaussian. This distribution on the hypersphere is exponentially larger in volume than the Gaussian according to the dimension. It is easy to use because it is close to the ordinary Gaussian in terms of formulation, and we can straightforwardly implement the algorithm on top of the naive VAE.
They also proposed the Will-It-Move test, where they fine-tune the parameters of the VAE to improve OOD detection performance further.
Experimental results on multiple public datasets demonstrate that the proposed method performs better than existing methods.

**Summary Of The Review:**

The proposed method is reasonable, and the experiments are good.
However, the comparison is only with the anomaly detection methods, and there is no comparison with the VAE variants that are proposed with the same motivation as the proposed method.

---

> ### Author Response · Authors · 2022-11-14
> **Response to Reviewer T18J**
>
> This is an excellent summary of the main points of the paper and you have clearly looked at the paper carefully. Thank you for the careful review! We have implemented the many suggestions in your review. Below we respond to the individual points from the review:
>
> **Re: Computing $\mu^{\star}$**
> $\mu^{\star}$ is the argmin of the KLD of the tilted Gaussian given explicitly in eqn (5). This explicit formula in eqn(5) involves the generalized Laguerre polynomial, but is otherwise quite a reasonable expression. In practice, we used gradient descent on eqn(5) to find $\mu^{\star}$ (see Figure 5 for the results). A comment about this was added to the appendix.
>
> **Re: Caption in Table 1**
> We reformatted this to make the caption more closely match the table. If you can suggest other ways to make this caption clearer, please let us know.
>
> **Re: Figure 3**
> We added some clarification to the figure, and it is referenced at the end of Section 4.1.
>
> **Re: The use of OOD samples in the Will-It-Move task:**
> Upon reading the reviewer’s comment “Although the WIM test is reasonable for their purpose, it uses samples from both the training and the OOD distribution.”, we realized that our explanation of the WIM test was not as clear as we had thought. The Will-It-Move task only needs two ingredients: samples of the original in-distribution task and unknown samples which are to be tested. An alternative set of samples which are a priori known to be OOD samples, as suggested by the reviewers comment, is not needed. We believe this confusion may have arisen from Figure 4 which shows “Test ID” and "Test OOD” points separately, but importantly this labeling is not needed in the running of the algorithm. (Instead, the labeling here serves only to showcase visually how well the WIM task is able to separate out these points on samples which the viewer knows the true identity of.) We have added a small remark in the caption of Figure 4 to make this clear.
>
> **Re: Hyperspherical VAE:**
> We have added a sentence to the related works section to clarify the difference between the hyperspherical VAE and our tilted prior. The hyperspherical uses a VMF distribution which is defined on the sphere, and requires the use of a VMF distributed encoder. In contrast, our tilted prior is concentrated around the hypersphere which allows the standard normal distribution to be used as the encoder distribution.

---

> > ### Comment · Reviewer_T18J · 2022-12-08
> > **Thank you**
> >
> > After reading the response and changes, I keep my score unchanged.
> > The description "an unknown data set U that has samples from both the training and the OOD distribution" is still confusing to me.
> > The comment "unknown samples which are to be tested" in the response is much easier to understand.
> > It would be better to explicitly mention that the unknown data set is test samples which consist of in-distribution and OOD samples and are independent of training samples.
> > I recommend that the authors further improve this description in the camera-ready version.

---

### Author Response · Authors · 2022-11-14
**Overall Response to Reviewers (detailed responses under each review)**

We would like to thank the reviewers for their thoughtful comments on our manuscript.  In some cases, their comments gave us valuable suggestions on how to improve the document which we implemented in the revised manuscript.  In other cases, they alerted us to the fact that some of our explanations were not as clear as they might have been; so, we improved our descriptions of what we did and why. In a few cases, they suggested some excellent avenues for expansion which we felt were beyond the scope of the current work; we would like to investigate these avenues in the future.  In all, we believe the document has been much improved by the revisions that we made in response to the reviewers’ comments, and have acknowledged this in the document.

A detailed list of changes is provided below in the individual responses to the reviewers. The rebuttal revision also has all new additions based on the reviewers comments highlighted in blue text so the reviewers can easily see the changes we made.

---

### Decision · Program_Chairs · 2023-01-20

**Decision:**

Accept: poster

**Justification For Why Not Higher Score:**

There are reservations about the issues related to the clarity of writing in many parts of the paper.

**Justification For Why Not Lower Score:**

The paper passes the acceptance bar of the conference.

**Metareview: Summary, Strengths And Weaknesses:**

This paper addresses the problem of KL collapse in variational autoencoder (VAE) by introducing a new prior for the latent vector called tilted Gaussian. Compared to the standard Gaussian prior, the volume of the tilted Gaussian on the hypersphere is exponentially larger than that of the Gaussian in high dimension. Furthermore, the paper also proposes a new paradigm for training OOD detection called "Will it Move". The experimental results are promising.

Overall, there is a general consensus among the expert reviewers that "technical novelty is high, ideas appear sound, experiments are convincing" with some reservations on issues related to the clarity and empirical comparisons to existing work. As a result, I recommend this paper (in the form that incorporates reviewers' feedback) for publication as a poster at ICLR 2023.

**Note From Pc:**

if the above contains the word "oral" or "spotlight" please see: "oral" presentation means -> notable-top-5% and "spotlight" means -> notable-top-25%. As stated in our emails, we are disassociating presentation type from AC recommendations

**Summary Of Ac-Reviewer Meeting:**

N/A